# Clinical application of tumour-in-normal contamination assessment from whole genome sequencing

Jonathan Mitchell[1,8], Salvatore Milite[2,3,8], Jack Bartram[4], Susan Walker[1], Nadezda Volkova[1], Olena Yavorska[1], Magdalena Zarowiecki[1], Jane Chalker[5], Rebecca Thomas[4], Luca Vago[6], Alona Sosinsky[1,9] ✉ & Giulio Caravagna[3,7,9] ✉

The unexpected contamination of normal samples with tumour cells reduces variant detection sensitivity, compromising downstream analyses in canonical tumour-normal analyses. Leveraging whole-genome sequencing data available at Genomics England, we develop a tool for normal sample contamination assessment, which we validate in silico and against minimal residual disease testing. From a systematic review of 771 patients with haematological malignancies and sarcomas, we find contamination across a range of cancer clinical indications and DNA sources, with highest prevalence in saliva samples from acute myeloid leukaemia patients, and sorted CD3+ T-cells from myeloproliferative neoplasms. Further exploration reveals 108 hotspot mutations in genes associated with haematological cancers at risk of being subtracted by standard variant calling pipelines. Our work highlights the importance of contamination assessment for accurate somatic variants detection in research and clinical settings, especially with large-scale sequencing projects being utilised to deliver accurate data from which to make clinical decisions for patient care.

In popular experimental designs for cancer bulk DNA sequencing— whole-exome sequencing (WES) or whole-genome sequencing (WGS) —individual tumour samples are matched with a reference "normal" sample from the same patient, usually obtained from peripheral blood, saliva, or a skin biopsy[1,2]. Analysis is performed by first detecting variants with respect to the human reference genome in both normal and tumour samples, followed by subtraction of the patient-specific variants from the tumour to select variants that are private to the tumour sample[3]. However, this experimental design is only effective if the matched normal sample is free of contaminating tumour cells, an assumption not often emphasised (Fig. 1a, b)[4,5]. When DNA derived from the normal sample is contaminated by tumour DNA, standard bioinformatics pipelines can mistakenly subtract genuine somatic mutations from the set of mutations identified in the tumour sample due to evidence for a mutation being present in the normal sample, resulting in a reduction in sensitivity for true somatic mutations (i.e. higher number of false negatives). Based on the clonal evolution model[6], the probability of false negative increases with the fraction of cells harbouring the mutation (due to a higher likelihood that the variant will be present in the sequencing data for the normal sample),

---

[1]Genomics England, London, UK. [2]Computational Biology Research Centre, Human Technopole, Milan, Italy. [3]Cancer Data Science Laboratory, Department of Mathematics, Informatics and Geosciences, University of Trieste, Trieste, Italy. [4]Department of Haematology, Great Ormond Street Hospital for Children, London, UK. [5]Specialist Integrated Haematological Malignancy Diagnostic Service - Acquired Genomics, Great Ormond Street Hospital for Children, London, UK. [6]Research Unit of Immunogenetics, Leukemia Genomics and Immunobiology, IRCCS Hospital San Raffaele, Milan, Italy. [7]Centre for Evolution and Cancer, The Institute of Cancer Research, London, UK. [8]These authors contributed equally: Jonathan Mitchell, Salvatore Milite. [9]These authors jointly supervised this work: Alona Sosinsky, Giulio Caravagna. ✉e-mail: alona.sosinsky@genomicsengland.co.uk; gcaravagna@units.it

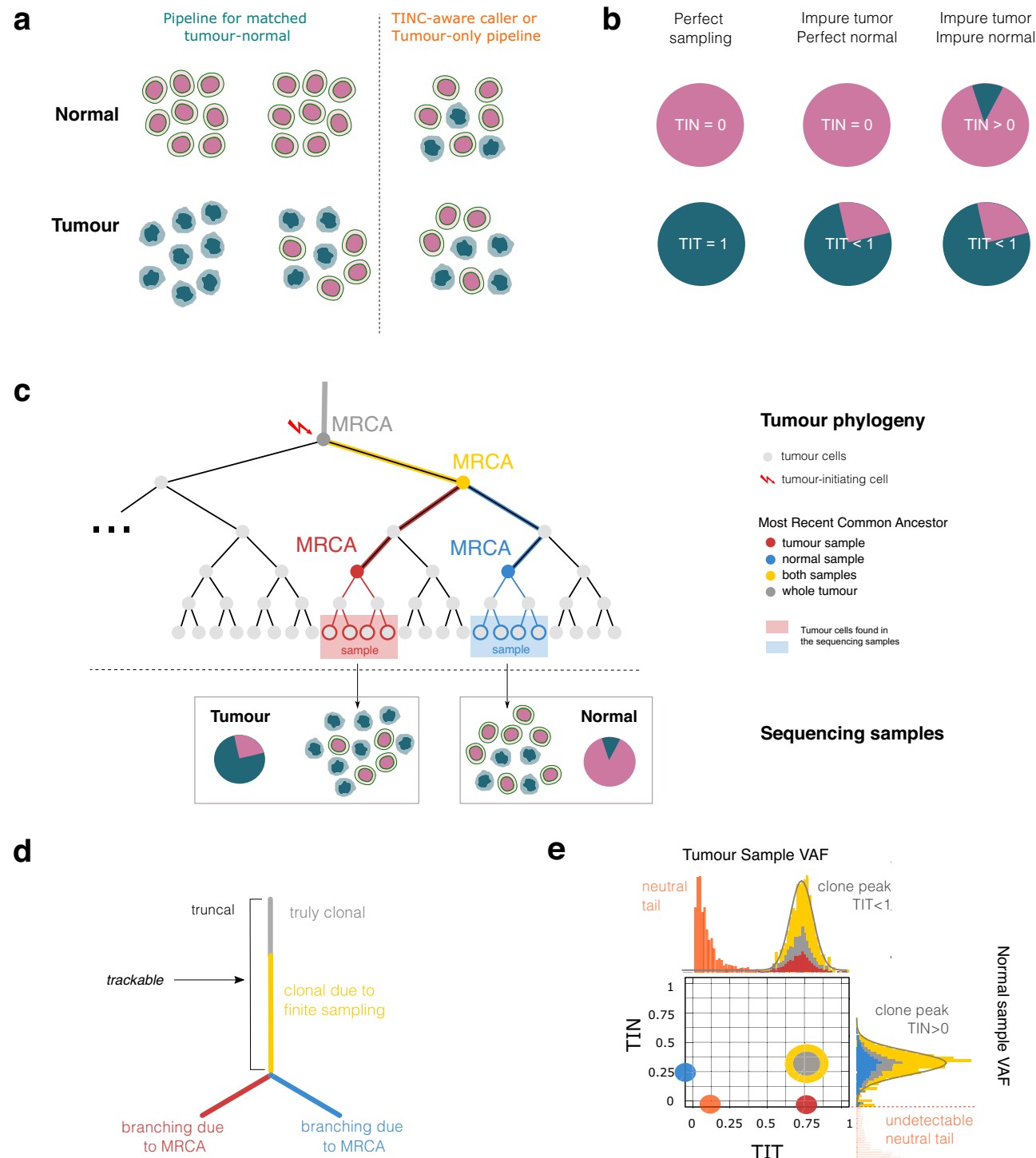

resulting in a bias towards erroneous subtraction of somatic variants at high allele frequency in the tumour. High allele frequency variants are of the greatest importance because they trigger tumour formation and determined the subsequent clonal evolution patterns[7].

More recently, in order to adapt the canonical bioinformatics somatic analysis approach to accommodate tumour in normal (TIN) contamination, a model for normal contamination was introduced into somatic variant calling algorithms[4]. However, these tools require an estimate of the TIN contamination level as an input parameter, and only support low contamination levels. Computationally, only one tool is available to assess tumour contamination of normal samples[8]. At high levels of TIN, however, the only alternative to a canonical tumour-matched-normal design is currently a tumour-only pipeline, using population germline frequency databases to filter likely germline variants from the set of putative somatic mutations. This approach attempts to minimise the rate of false negatives, a fundamental requirement for clinical reporting where the failure to identify actionable somatic mutations can be detrimental to patient care, but leaves many ultra-rare germline variants of indeterminate origin.

Here we present TINC, a computational method to assess the level of TIN contamination applicable for tumour and matched normal pairs, leveraging state-of-the-art tools for measuring clonal evolution from WGS. We demonstrate the performance of TINC with simulated data and by comparison with orthogonal minimal residual disease

**Fig. 1 | TINC method. a** Cellular composition of a bulk tumour and normal sample (e.g. peripheral blood, saliva, or skin biopsy). Ideally, there would be no cross-contamination between tumour and normal samples (pink and teal cells show perfect separation). In reality, all tumour samples contain normal cells. For a paired analysis, challenges in somatic variant detection arise when the normal sample is contaminated with tumour cells, resulting in subtraction of true somatic variants and a decrease in variant detection sensitivity. **b** The level of contamination of a bulk sample can be defined as the fraction of tumour cells in the sample. With perfect sampling tumour purity (TIT score) equals 1, and tumour-in-normal contamination (TIN score) equals 0; for most real-life samples, TIT < 1. TIN > 0 in normal sample with tumour contamination. **c** Tumour cell phylogeny showing cell divisions as a tree representing the evolutionary relationship between sampled tumour cells. Colours represent distinct Most Recent Common Ancestors (MRCAs) of the tumour cells, according to sampling. With TIN contamination, phylogenetically related tumour cells are found in both samples (yellow and grey trunk). Tumour cells found in the tumour and the normal carry common as well as private mutations (red for the tumour and blue for the normal). TIN and TIT are determined using the mutations accrued up to the yellow MRCA, an ancestral cell common to the tumour cells present in both samples. **d** Summary phylogenetic tree for the cell divisions in (**c**) shows a branching effect that describes a lineage division and spatial sampling bias. **e** Expected cell fraction distribution for tumour cells in tumour and normal samples carrying ancestral (yellow and grey) and private (blue and red) mutations for a case with TIT = 75% and TIN = 25%. Somatic mutations common to tumour cells found in both samples including the key tumour truncal driver mutations, which are frequently subtracted in tumour-normal analysis, are the yellow and grey cluster. Mutations only found in the tumour cells within the normal sample (shown in blue) have no read support in the tumour and are not considered by standard somatic variant callers.

(MRD) test data for 70 leukaemia patients. By applying TINC to sequencing data from 771 participants in the Genomics England 100,000 Genome Project[28], including 617 patients with haematological malignancies, we detect tumour contamination in normal samples from different germline sources across several tumour types. Assessment of sample quality is essential in clinical reporting where treatment decisions are based on genomic data. The level of contamination predicted by TINC provides reassurance for clinicians and clinical scientists for the accuracy of variant detection in the somatic analysis, and can highlight the risk that clinically relevant variants may not have been reliably detected. Therefore, we propose that TIN is an essential metric for clinical analysis and reporting of WGS data.

## Results

### The TINC method

We have developed TINC, an approach for tumour-in-normal contamination assessment, leveraging the concept of tumour clonal evolution. It is freely available as an open-source R package (Data Availability). TINC uses the variant allele frequencies (VAFs) of somatic single nucleotide variants (SNVs) detected in tumour and normal samples to identify clonal somatic mutations (i.e. those detected in all tumour cells sampled) and evaluate their level in the normal sample. From the observed VAFs, TINC determines scores for the percentage of tumour cells in the tumour (usually referred to as tumour purity) and in the normal sample, which are called Tumour in Tumour (TIT) and Tumour in Normal (TIN) scores respectively. These scores can be expressed as units of read fractions (i.e., percentage of reads in the sequencing data for the normal sample that are derived from the tumour).

TINC first identifies high-confidence clonal somatic mutations in the tumour sample (grey, yellow and red lineages in Fig. 1c–e), which are then used to estimate tumour purity (TIT score). Support for these clonal variants is then assessed in the normal sample to determine the level of tumour-in-normal contamination (TIN score, Fig. 1e). Specifically, TINC targets the ancestor of the tumour cells found in both the tumour and normal samples, i.e., the most recent common ancestor (MRCA) of the sequenced cells (yellow and grey lineages in Fig. 1c–e) which can not be directly sampled or sequenced. Due to sampling differences, private mutations are anticipated in the different tumour cell lineages present in the tumour and the normal samples (red and blue lineages in Fig. 1c–e). Note that clonal somatic mutations can be mistakenly labelled as private tumour mutations due to the difference in sequencing read depth between tumour (about 100x) and normal (about 30x), as well as sequencing noise that obscure support for clonal somatic mutations in normal samples. Therefore, the assessment of clonal mutations in the normal sample is impacted by the tumour architecture and data quality at variant sites. Building from the expected tumour architecture, we are able to model the anticipated data distribution in both samples for somatic variants of the various cell lineages (Fig. 1e).

Using TINC, clonal mutations are identified with the MOBSTER machine learning model for subclonal deconvolution from WGS[9]. MOBSTER integrates population genetics and machine learning to cluster somatic variants based on their VAF, decoupling clones that undergo positive selection from neutral mutations. Read counts for high-confidence clonal somatic mutations identified by MOBSTER are then fitted to a Binomial mixture in the matched normal sample. From the analysis of the tumour and normal samples, TINC obtains the information to compute TIT and TIN scores (Online Methods).

TINC can also utilise allele-specific tumour Copy Number Alteration (CNA) calls to retain SNVs in a subset of genomic intervals with the copy number state (e.g. heterozygous diploid or tetraploid) spanning the largest proportion of the tumour genome. By incorporating CNAs in the logic, TINC normalises the observed VAFs in the tumour sample for chromosome copy number and therefore is resilient to confounding effects of CNAs. Incorporation of copy number data is only performed for tumours for which the most extensive copy number state is one of 1:0 (loss of heterozygosity, LOH), 1:1 (heterozygous diploid), 2:0 (copy-neutral LOH), 2:1 (triploid) or 2:2 (tetraploid genome-doubled), representing the majority of copy number states observed for cancer genomes[10]. For such cases, only SNVs residing within regions of the most prevalent copy number state are used in TIN estimation. The SNV-only analysis (i.e. not incorporating CNA data) requires that the tumour genome does not harbour a high number of CNAs as otherwise the value of VAF is cofounded by copy number variations[10].

### In silico validation of TINC performance

The performance of TINC was assessed using WGS data generated by Genomics England for participants recruited as a part of 100,000 Genomes Project[28]. Synthetic test data were generated by artificially contaminating normal BAM files with sequencing reads sampled from the corresponding tumour BAM file (Methods). Variant calling of the synthetic samples was performed using the Genomics England bioinformatics pipeline and the generated SNV and CNA calls analysed with the TINC package (Fig. 2a).

Synthetic TINC test data were generated using high-quality WGS data derived from seven patients diagnosed with haematological malignancies (acute lymphoid leukaemia, ALL; acute myeloid leukaemia, AML; multiple myeloma, MM) and five patients diagnosed with lung cancer (adenocarcinoma or squamous cell carcinoma), for whom the normal sample sequenced was not affected by tumour contamination. In total, artificially contaminated WGS data were generated for thirty-nine haematological tumour-normal pairs and thirty lung cancer tumour-normal pairs, with a range of TIN contamination from 0% to 25%.

TINC successfully estimated the correct level of contamination for the majority of synthetic WGS samples, in both haematological and lung cancers (Fig. 2b, c, $R^2 = 0.95$; $p < 2.2 \times 10^{-16}$ and $R^2 = 0.85$; $p < 3.3 \times 10^{-13}$, correspondingly). These data demonstrated the benefit

**a**

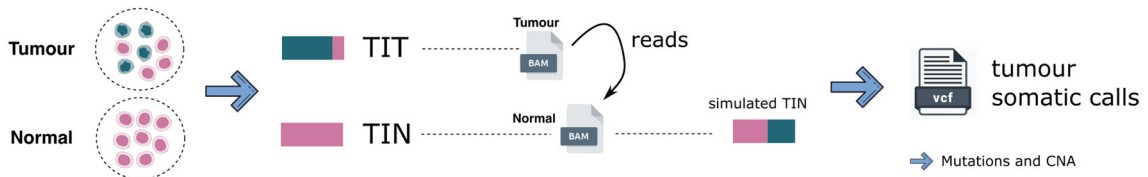

**b** Hematological (in–silico contamination)
n = 39 samples from 7 patients

$R^2 = 0.94$, $p < 2.2e{-}16$

**c** Lung adenocarcinomas (in–silico contamination)
n = 30 samples from 5 patients

$R^2 = 0.85$, $p = 3.3e{-}13$

Clonal SNVs (relative %)    CNA Copy state ● 1:1  ▲ 2:2
0.4 0.6 0.8 1.0

Clonal SNVs (relative %)    CNA Copy state ● 1:1  ▲ 2:1  ■ 2:2
0.5 0.6 0.7 0.8 0.9 1.0

**d** TINC and DeTin on hematological cancers

$R^2 = 0.97$, $p < 2.2e{-}16$, (TINC)

$R^2 = 0.9$, $p = 1.3e{-}12$, (DeTin)

TIN/TIT (RF)
Relative tumour DNA abundance in normal (RF)

CNA Copy state ● 1:1

**e** TINC and DeTin on lung cancers

$R^2 = 0.93$, $p = 2.2e{-}14$, (TINC)

$R^2 = 0.91$, $p = 3.8e{-}13$, (DeTin)

TIN/TIT (RF)
Relative tumour DNA abundance in normal (RF)

CNA Copy state ● 1:1  ▲ 2:1  ■ 2:2

of incorporating CNAs, particularly for cancers with high chromosomal instability, where failing to account for CNAs can result in overestimation of TIN contamination (Supplementary Fig. S1a, b). Notably, the lung cancer samples used here showed a high level of copy number variation, with an average of 78% of the genome covered by CNAs (Supplementary Fig. S2). For a number of cases, the level of contamination was underestimated at higher levels of TIN contamination. This decrease in performance was attributed to the impact of high levels of TIN contamination on somatic variant detection. At high levels of TIN contamination, the somatic variants with the highest VAF

in the normal sample are likely to be subtracted from the somatic variant set. This results in an underestimate of the TIN score since the variant set used for the estimation of TIN score is biased towards variants with lower VAF in the normal sample. This effect is illustrated in Figs. 2b, c by the gradient colour representing the decreasing fraction of clonal mutations used in TIN score estimation for increasing levels of TIN contamination. This limitation is unlikely to impact clinical reporting in practice, as the effect is only significant for samples with a level of TIN far higher than would be considered acceptable for clinical-grade analyses. At very high levels of TIN contamination the

**Fig. 2 | In silico validation of TINC performance. a** Generation of test data by in silico contamination of patient WGS datasets. A range of TIN levels were generated from tumour and normal BAM files, injecting tumour reads in the normal BAM to achieve a desired level of TIN contamination. Somatic variant calling of small variants and CNAs was performed by pairing the original tumour BAM with the in silico contaminated normal, and the resulting calls used for TINC analysis. **b** Performance of TINC with the in silico contaminated haematological cancer samples. The scatter plot compares the expected TIN contamination (based on in silico contamination) to TINC estimates. Both axes report the score in read fractions for the tumour (RF). Each point is coloured by the percentage of clonal mutations used by TINC, relative to the original uncontaminated sample. The fraction of clonal mutations decreases with increasing contamination, due to the limitations of variant callers that fail to report genuine somatic variants (false negatives). With few clonal mutations, identifying clonal peaks is more difficult; in this case clonal variants are also biased

towards those with lower support in the normal sample. Line fits were performed by linear regression (tests with Pearson method with two-sided p-value and squared correlation coefficient). **c** Performance of TINC with lung cancer samples contaminated in silico. The same information available in (**b**) is provided. These tumours have a higher fraction of CNAs compared with haematological cancers that are represented by triangles and squares. Fits and tests are as in (**b**). **d**, **e** Performance of DeTiN and TINC on the haematological and lung cancer samples shown in (**b**) and (**c**). Consistent with the definition of DeTin, the relative tumour DNA abundance in the normal and tumour samples is shown on the x-axis. This plot is restricted to cases with a maximum ratio of 20%, which includes samples within the anticipated contamination range for use in clinical reporting (full plot, Supplementary Fig. S1). The y-axis shows the ratio between TIN and TIT scores returned by the two tools. Fits and tests are as in (**b**). Source data are provided as a Source Data file.

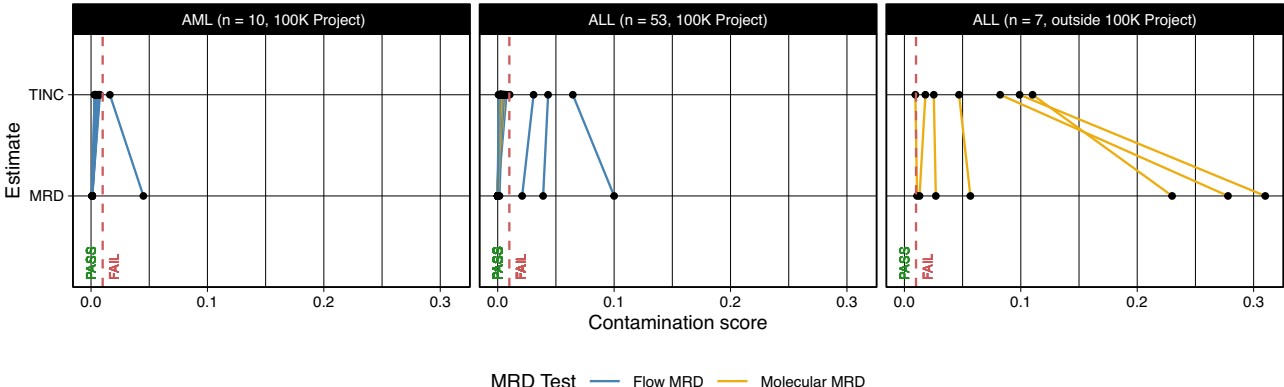

Fig. 3 | **Validation of TINC by comparison with orthogonal test data generated either by flow cytometry or molecular Minimal Residual Disease (MRD) test.** Here 63 patients are recruited through the 100,000 Genomes Project (10 AMLs and 53 ALLs), while 7 ALLs are not (criteria for project enrolment reported in Supplementary Table 1). The threshold for TIN contamination (>1% TIN) is shown with the dashed vertical line. Source data are provided as a Source Data file.

majority of clonal somatic variants are subtracted during the somatic variant calling, leaving only subclonal variants which can lead to a strong underestimation of TIT score. However, the implementation of appropriate quality control thresholds can be used to prevent this scenario resulting in high TIN contamination going undetected.

To further assess TINC's performance, we compared outcomes with the only alternative method available to assess tumour contamination of normal samples, DeTiN[8]. DeTiN was found to perform well with WGS data, generating results similar to TINC ($R^2$ scores exceeding 0.9 for both cohorts, note that $R^2$ score is calculated for TIN/TIT ratio) (Fig. 2d, e, and Supplementary Fig. S1). However, TINC is more flexible as it can process copy number segments regardless of which CNA caller has been used. In contrast, DeTiN requires mean allele fraction of minor parental allele at each segment and two centred segment copy ratio data. In addition, TINC generates absolute TIN and TIT values which are directly compatible with orthogonal minimal residual disease (MRD) tests used in clinical practice. The DeTIN ratio can be converted to absolute TIN value if a separate calculation for the tumour purity estimate (TIT score) is performed using copy number calling data and used for calibration. However, TINC is able to estimate absolute TIN value without resorting to any additional external inputs.

### Experimental validation of TINC performance
To further validate TINC, contamination estimates for a cohort of participants in the 100,000 Genomes Project with haematological cancers (see Contamination analysis of clinical samples) were compared either with molecular Minimal Residual Disease (MRD) test using

real-time PCR assessment of rearranged immunoglobulin/T-cell receptor genes or flow cytometry test for leukaemia-associated immunophenotype (Fig. 3; FACS sequential gating strategies in Supplementary Fig. S3). For 53 ALL and 10 AML samples assessed (Fig. 3), the estimates from TINC and MRD were consistent, with the same four of 63 samples considered as being contaminated (i.e. TIN contamination >1%). In order to extend experimental validation to additional samples with TIN, we included in our validation cohort seven additional ALL samples not meeting the 100,000 Genomes Project sample collection criteria (Fig. 3). Consistent with observations from the in silico validation experiment, the TIN value was underestimated when compared with experimental data but still very significant for samples with high TIN.

### TINC implementation in a high throughput bioinformatics pipeline
TINC has been implemented as an essential quality control step for contamination assessment in the high-throughput bioinformatics pipeline at Genomics England (Fig. 4a). Using variant call format (VCF) files generated in the somatic SNV and CNA detection components of the analysis pipeline, TINC classifies normal samples as "PASS", "FAIL" or "Cannot estimate TINC reliably", based on the level of TIN and TIT detected. For all analysed samples, TIT and TIN scores are presented as read-fractions (RF), i.e. the fraction of reads in a given sample originating from the tumour. Samples with TIN >1% are classified as "FAIL" indicating an alternative analysis not reliant on the matched normal sample should be conducted (for example, using an unmatched normal sample from another individual). As deconvolution of the VAF distribution is typically unreliable for tumour samples with low tumour

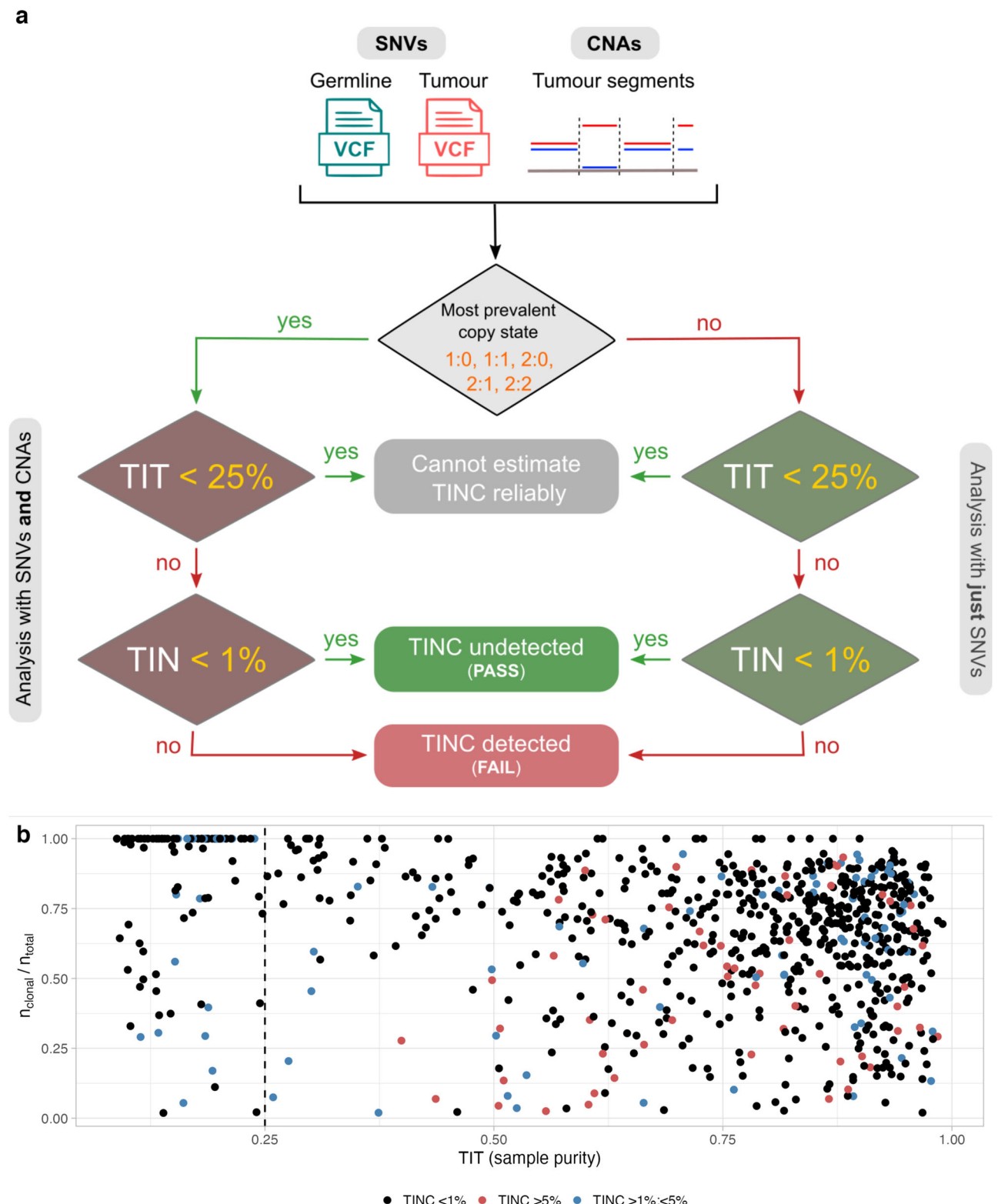

**a**

**b**

purity, samples with a TIT score estimated at <25% are considered as not eligible for TIN estimation (cluster of samples with clonal/total mutations ratio = 1 on Fig. 4b) and classified as "Can't estimate TIN reliably". This cutoff has been estimated using a cohort of 617 whole-genome samples (see Contamination analysis of clinical samples), and accounts for i) samples with genuinely low tumour purity and for ii) samples for which TIT is underestimated due to high TIN (i.e., extreme high TIN causing catastrophic subtraction of clonal somatic SNVs).

In order to mitigate the potential loss in variant calling sensitivity due to TIN contamination, in the Genomics England bioinformatics pipeline (Pipeline 2.0), small variants and structural variants in hae-matological samples with TIN >1% or samples where TIN can't be estimated are also analysed in a parallel pipeline, without subtraction using the patient's germline (tumour only). In this pipeline, filtering of contaminating germline variants is performed for variants with population allele frequency >0.01 in population databases, and

**Fig. 4 | TINC test implementation in Genomics England pipeline. a** Somatic SNVs are used in TIN assessment; by default all variants are used. If run with CNA integration, only SNVs mapping to the most prevalent copy state are used. The supported copy states are 1:0 (loss of heterozygosity, LOH), 1:1 (heterozygous diploid), 2:0 (copy-neutral LOH), 2:1 (triploid) or 2:2 (tetraploid genome-doubled) TIN contamination is estimated for samples with tumour purity (TIT score) >25%. Samples that can be analysed are assigned a TIN score, which can be converted into tumour read fractions (RF) detected in the normal sample, and used to determine a final status for the presence or absence of contamination. The threshold implemented at Genomics England to determine PASS status (TIN contamination undetected) versus FAIL (TIN contamination detected), is set to 1% RF. **b** Scatter plot reporting the ratio between the number of clonal mutations over total mutational burden, against estimated sample purity (TIT) for 617 WGS samples of haematological cancers. When clonal/total mutations ratio = 1, TINC did not separate clonal somatic variants from subclonal variants and TIN estimates are less reliable. The majority of samples with ratio = 1 are clustered with TIT score < 25%. The colour of each point represents the sample contamination as estimated by our method; the vertical dashed line represents the 25% purity cutoff for TINC analysis adopted in Genomics England. Further details on this cohort and contamination assessment are shown in Fig. 6. Source data are provided as a Source Data file.

potential sequencing artefacts using a Panel of Normals (PON) approach (Online Methods). The results of the two pipelines are subsequently merged and analysed together in the annotation and interpretation workflow (Fig. 5a). This hybrid configuration allows high-confidence somatic variants from the paired tumour-normal pipeline to be combined with variants of uncertain origin from tumour only pipeline, to ensure no reduction in overall sensitivity. The filtering thresholds have been optimised to reduce the number of non-somatic variants in clinically relevant genes returned using the tumour-only pipeline without compromising sensitivity for identifying true somatic variants selected from a manually curated set of 65 haematological samples (Fig. 5b, c). In order to demonstrate sensitivity for somatic variant detection despite increasingly high levels of TIN contamination, we compared the sensitivity of the paired tumour-normal pipeline (Fig. 5d) with the tumour-only pipeline (Fig. 5e).

## Contamination analysis of clinical samples

To assess the clinical impact of implementing TINC in a bioinformatics pipeline, TINC was used to analyse 771 tumour-normal pairs from participants in the 100,000 Genomes Project with either haematological cancer (n = 617) or sarcoma (n = 154). All samples were re-analysed through the Genomics England Pipeline 2.0 (Online Methods). Since TIN contamination is not expected to be a frequent occurrence for sarcoma samples, these samples were included as a control group for comparison. Normal DNA used for WGS was derived from blood, cultured fibroblast, saliva or skin biopsy samples. TIN scores were determined only for the samples with TIT score above 25%.

The haematological malignancy samples covered a wide range of the most common clinical indications: AML (n = 168), chronic lymphocytic leukaemia (CLL, n = 158), MM (n = 87), ALL (n = 90), myeloproliferative neoplasm (MPN, n = 58), chronic myeloid leukaemia (CML, n = 57), diffuse large B-cell lymphoma (DLCBL, n = 23), low and moderate grade non-Hodgkin B-cell lymphoma (Low/mid grade NHL, n = 20) and high-risk myelodysplastic syndrome (High-risk MDS, n = 20). Normal sample collection was performed according to criteria established for the 100,000 Genomes Project (Supplementary Table 1).

Of the 771 cases assessed, CNA and SNV data were incorporated for 758 cases, with TIN score estimated using SNV data only (due to sample ploidy not matching TINC criteria) for 13 cases. The proportion of cases for which TIN contamination was observed (tumour read fraction in normal sample >1%) varied across the cancer subtypes (Fig. 6a–c). As expected, no normal samples derived from the cultured fibroblasts had TIN >1%. Notably, the two cancer subtypes with the highest fraction of contaminated samples were MPN and AML. 22 of 24 (91%) of sorted CD3+ T cell-derived normal samples for MPN cases and 43 of 114 (38%) saliva-derived normal samples for AML cases were found to be contaminated, consistent with previous reports[11–13]. It is worth noting that saliva samples were accepted as a normal sample in myeloid malignancies only if sufficient treatment has been given to remove all circulating myeloid cells from the peripheral blood, e.g. after administration of anthracycline chemotherapy in patients receiving intensive induction in AML.

In contrast, among the sarcoma samples, only 2% were found to be contaminated (4 out of 154, 3 out 4 with TINC below 2%). This can be explained by the fact that two of the contaminated normal samples for sarcoma patients were derived from fresh frozen muscle tissue, which can present a higher risk for contamination than blood in patients with solid cancers. In an additional group of ALL cases for whom the sample collection procedures did not meet the criteria specified for the 100,000 Genomes Project, 11 of 46 (24%) showed contamination. A full description of the samples tested and the TINC results are provided in Supplementary Figs. S4–7.

To assess the incidence of essential somatic variants being at risk of inappropriate subtraction during tumour-normal analysis, we computed the read-support in tumour and normal samples for hotspot mutations in the AML and MPN patient cohorts. We focused on genes with the highest prevalence of somatic mutations in haematopoietic and lymphoid tissue in COSMIC: *JAK2, FLT3, DNMT3A, TP53, KIT, NRAS* and *IDH2*, and defined hotspot mutations as those found in at least 100 samples in COSMIC. Of 108 high-confidence hotspot mutations with VAF in tumour >5% identified in the AML and MPN samples, 51 had a VAF >1% in the normal sample and 27 >5% (Supplementary Fig. S8). We found hotspot mutations with a VAF >10% in the normal samples in 6 AML cases with *DNMT3A p.R882X*, 5 with *IDH2 p.R140Q*, 3 with *JAK2 p.V617F and* 1 with *TP53 p.R273X, IDH2 p.R172K and NRAS p.Q61X*, and in 4 MPN cases with *JAK2 p.V617F*.

This set of mutations overlaps with those commonly found in clonal haematopoiesis of indeterminate potential (CHIP)[14,15], the presence of a pre-cancerous clonally expanded hematopoietic stem cell population, caused by a somatic mutation that can, potentially, cause malignant transformation. In order to investigate further the relationship between CHIP and TIN contamination, we scanned the sequencing data for the normal samples in the cohort of 168 AML patients for the presence of 168 point mutations previously reported in genes linked with CHIP and myeloid malignancies (*IDH2, PRPF8, PPM1D, SRSF2, TP53, GNB1, ASXL1, GNAS, RUNX1, SF3B1, DNMT3A, MYD88, CCND3, TET2* and *JAK2*)[15]. We observed a weak correlation between TIN score and VAF for the CHIP mutations in normal samples (Supplementary Fig. S9) that can reflect complex phylogenetic relationships between the hematopoietic and the AML clones. Overall we demonstrated that TINC was able to flag normal samples with recurrent CHIP mutations, and to trigger a hybrid pipeline that includes tumour-only analysis in order to report the true extent of tumour mutations (including CHIP).

The availability of WGS allows us to examine the extent of contamination for all tumour clonal mutations, going beyond hotspot ones (which are usually clonal). In Fig. 7 and Supplementary Figs. S10–S11 we report two example cases of AML patients from the 100,000 Genomes Project with tumour in normal contamination. In the first case, 982 diploid SNVs are analysed by the TINC test (see Fig. 7a–c for VAF distribution in tumour and normal samples), of which 378 are identified as clonal mutations by deconvolution analysis (teal dots on Fig. 7e). Deconvolution analysis identified two clusters of subclonal mutations in addition to the clonal cluster (green and blue peaks in Fig. 7d, top). The allele fraction in the normal sample for the clonal variants peaks at ~8% (Fig. 7d, bottom), indicating that approximately 16% of cells sampled in the normal sample are of tumour origin (Fig. 7f). In this case, the analysis performed during the

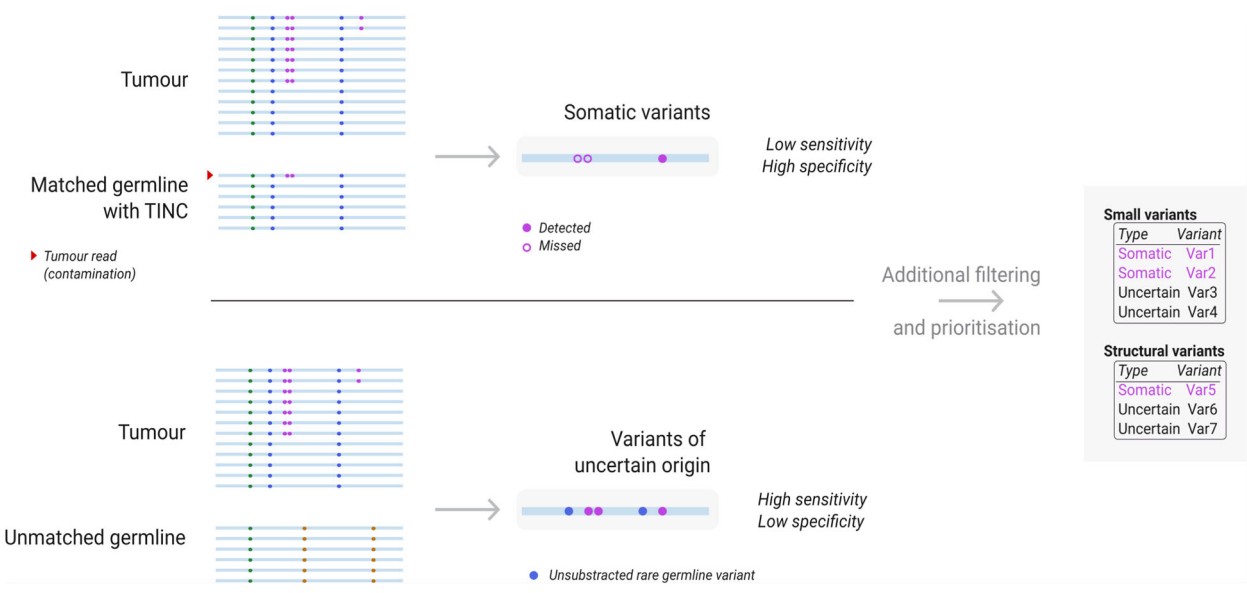

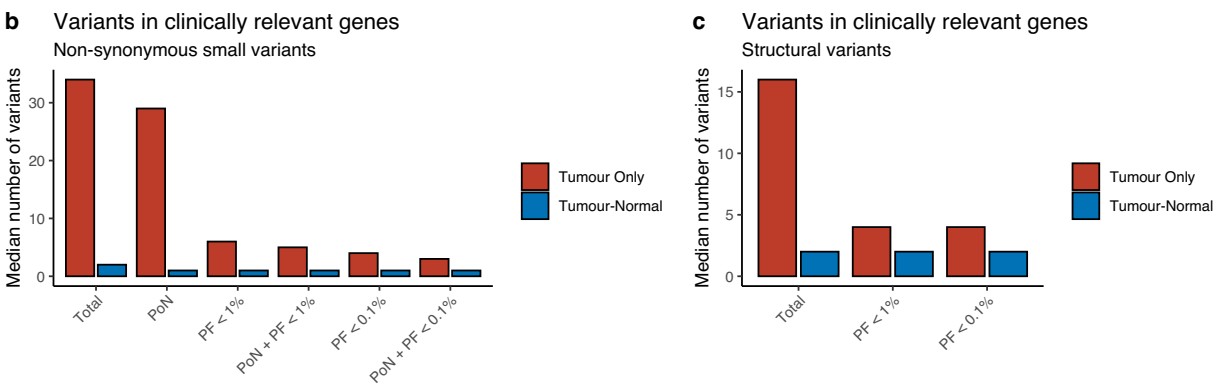

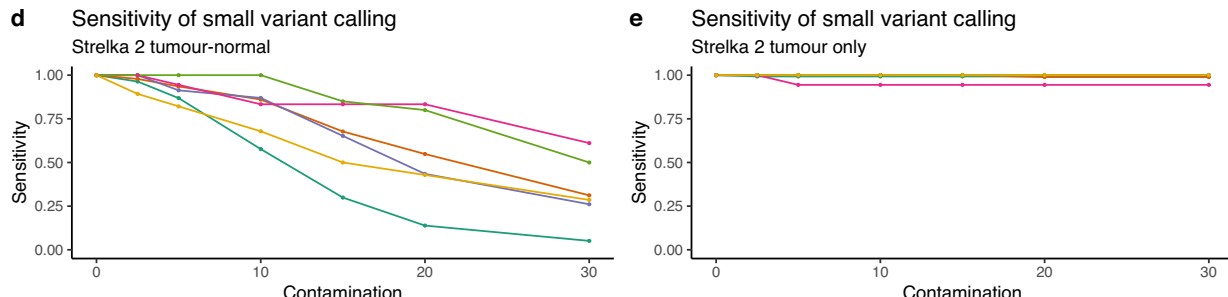

**Fig. 5 | Hybrid variant calling pipeline for processing of samples with TIN contamination. a** Graphical representation of the pipeline that combines outputs of paired tumour-normal run with high specificity and reduced sensitivity due to TIN contamination and tumour only run (unmatched normal sample is used to satisfy input requirements) with high sensitivity and low specificity due to unsubtracted rare germline variants. **b**, **c** Extensive filtering is therefore implemented to reduce the number of variants in clinically relevant genes reported from tumour only workflow. Panel of Normals (PoN) is applied to SNVs to reduce the number of false positive findings due to sequencing artefacts. Population Frequency (PF) filter is applied to reduce the number of common germline variants in tumour only run. Filtering cut-offs are optimised for improving specificity without compromising sensitivity. Application of these two filters significantly reduces the number of SNVs (**b**) and SVs (**c**) that require clinical review. **d**, **e** Sensitivity of SNV calling for samples from Fig. 2b with standard paired tumour-normal analysis (**d**) and with tumour-only pipeline (**e**). Source data are provided as a Source Data file.

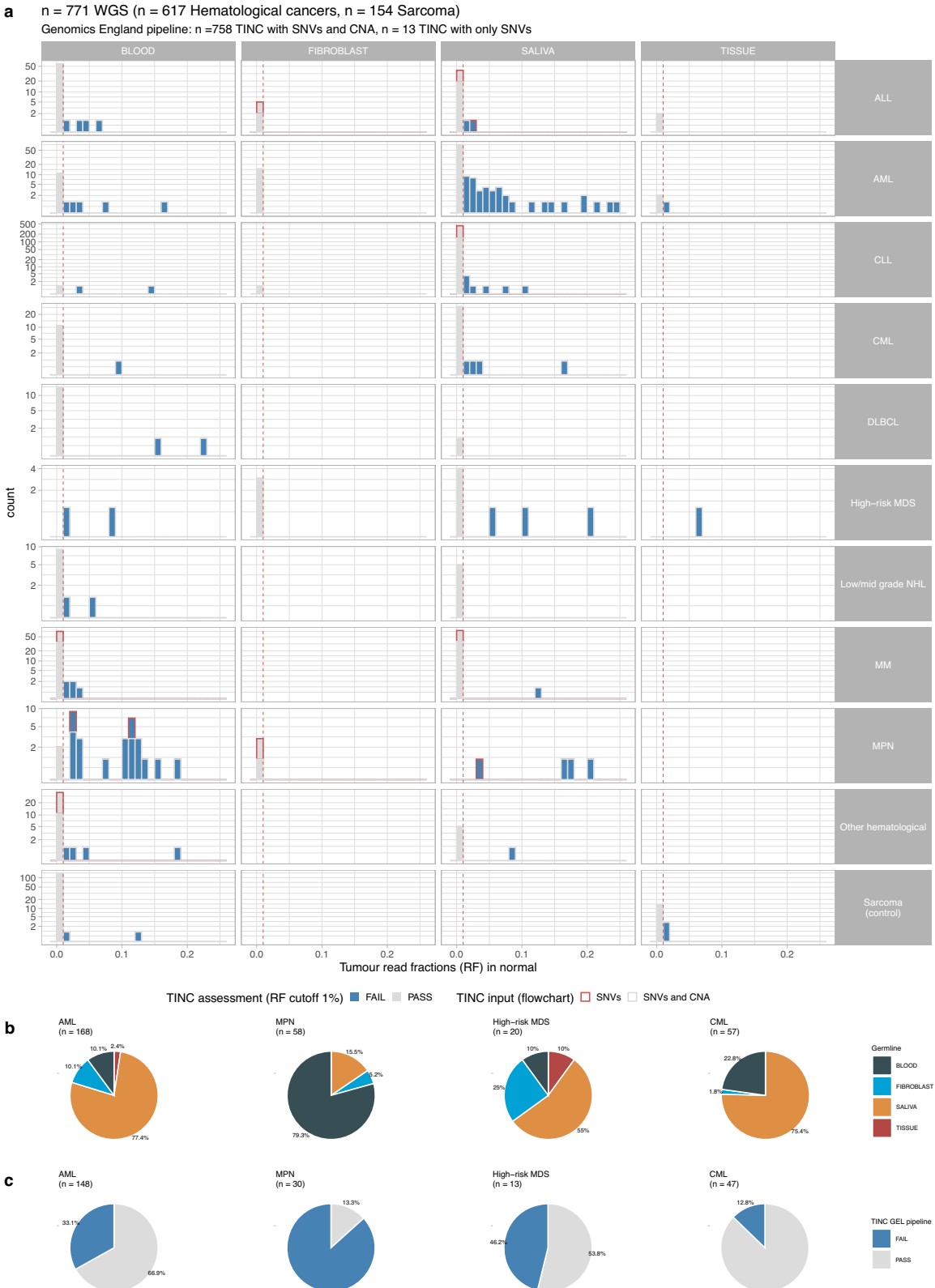

**a**  n = 771 WGS (n = 617 Hematological cancers, n = 154 Sarcoma)
Genomics England pipeline: n =758 TINC with SNVs and CNA, n = 13 TINC with only SNVs

100,000 Genomes Project (Genomics England pipeline 1.0, Online Methods) failed to detect a driver somatic hotspot variant in the *JAK2* gene (ENST00000381652:p.V617F) due to TIN contamination (variant supported by 50 out of 110 paired reads in the tumour, and 4 out of 34 in the normal; Fig. 7a). Additionally, a frameshift deletion in *TP53* (ENST00000269305:c.594delA) was observed at 28% VAF in the normal sample raising uncertainty as to whether this variant is germline or

somatic (variant supported by 41 out of 90 paired reads in the tumour, and 6 out of 21 in the normal), a classification that is important for clinical reporting. Additionally, if somatic, as this variant has a higher VAF in the normal sample than all other somatic variants (Fig. 7a) one can hypothesise that it is a marker of CHIP. In the second example case, 358 clonal SNVs are used to estimate TIN contamination, again at approximately 16% tumour cells in the normal sample (Fig. 7g-i).

**Fig. 6 | Application of TINC to the 100,000 Genomes Project dataset.**
**a** Distribution of the estimated level of tumour in normal contamination for 771 tumour-normal pairs derived from participants in the 100,000 Genomes Project (*n* = 617 haematological cancers, *n* = 154 sarcomas). Data are shown for haematological cancers of the subtypes: Acute Lymphoblastic Leukaemia (ALL), Acute Myeloid Leukaemia (AML), Chronic Lymphocytic Leukaemia (CLL), Chronic Myeloid Leukaemia (CML), Diffuse Large B-cell Lymphoma (DLCBL), High-risk Myelodysplastic Syndrome (High-risk MDS), Low and moderate grade Non-Hodgkin B-cell Lymphoma (Low/mid grade NHL), Multiple Myeloma (MM) and Myeloproliferative Neoplasm (MPN). Azure bars represent normal samples with TIN score >1%

expressed in read fractions, light grey bars l samples with score <1%. **b** Distribution of normal sample source for haematological cancers. The fraction of normal samples for which the DNA was derived from blood, saliva, fibroblasts or tissue samples is shown for haematological cancers of different subtypes (AML, MPN, High-risk MDS and CML). **c** The proportion of normal samples determined to have a PASS or FAIL status by TINC (1% read fraction threshold) is shown in light grey and azure respectively for AML, MPN, High-risk MDS and CML cancers. The proportion of cases that could not be analysed by Genomics England pipeline (tumour purity estimated to be below 25%) is shown in dark grey. Source data are provided as a Source Data file.

Analogous to somatic small variant calling, somatic structural variant calling also requires the subtraction of germline variants and therefore TIN can result in false negatives. In this case, a diagnostic somatic *PML-RARA* fusion was not detected due to TIN contamination (variant supported by 25 out of 150 paired reads in the tumour, and 6 out of 52 in the normal), which would affect patient diagnostic classification[16]. Further examples of the determination of TIN status using tumour-normal pairs from participants with MPNs are shown in Supplementary Figs. S12–S13.

## Discussion

Typical tumour-normal analyses utilise a normal sample to identify the patient's germline variants, which are then subtracted from the variants identified in the tumour to define the tumour-specific somatic mutations. This approach is successful when high-quality normal samples are available, but somatic variant detection is significantly impacted by poor-quality normal samples, particularly those in which there is contaminating tumour DNA. The incidence of contaminated normal samples has been understudied, thus some tumour types and normal source combinations may have an unforeseen prevalence of TIN contamination, making them unsuitable for a canonical tumour-normal analysis.

Our computational technology to assess TIN contamination exploits tumour evolutionary principles to quantify the proportion of contaminating tumour cells in a normal sample, using WGS data for paired tumour and normal samples. The score generated by TINC is simple to interpret and can resist confounding factors such as tumour CNAs and tumour sample purity. Furthermore, it can be computed automatically by integrating different data types, regardless of the variant calling methodologies used. The TIN score therefore allows an informed decision-making process to either proceed with variant interpretation and reporting, or whether alternative variant calling procedures are required, such as tumour-only analysis.

By applying TINC to WGS data from a large cohort of tumour-normal pairs, we have performed a thorough investigation of TIN contamination in the most common types of haematological cancers and sources of normal DNA. Haematological cancers were the ideal candidates for this assessment due to the natural spread of tumour cells in the bloodstream. Previous studies have demonstrated that saliva DNA from MPN patients can be positive for *JAK2 p.V617F* mutations[12,17]. A more recent study also reported contamination in saliva samples of MPN patients due to leucocyte presence in the oral mucosa, and suggested the use of CD3+ T cells as a source of normal sample[11]. Strikingly, we found that WGS data for normal DNA derived from sorted CD3+ T cells for MPN patients show clear signs of contamination and exhibit *JAK2* mutations with high read support.

We also found a high prevalence of TIN contamination in saliva samples for AML patients. Similar to our findings, other studies report the oral cavity presents the first clinical manifestations of leukaemia[18], with gingival infiltration of AML cells demonstrated by biopsy[19,20] and most commonly seen in acute monocytic leukaemia and acute myelomonocytic leukaemia[13,21].

Given the incidence of these tumour types and current standard practices for normal sample collection, findings with our TINC tool are crucial for improving best practice guidelines for sample handling and

highlight the importance of thorough quality control processes, particularly when genomic data are used to inform clinical decisions. Failure to identify TIN contamination increases the potential for false negative somatic variants, particularly those that occurred within the earliest stages of tumorigenesis and thus have the highest representation in the sequencing data. These mutations could determine disease course[7], stratify patients with respect to treatment response (McGranahan and Swanton 2017)[22], and inform targets for therapy (McGranahan et al. 2016)[23], so failure to detect them carries a high risk for clinical reporting, familial screening, and basic cancer research.

Reassuringly, our findings demonstrated the validity of specific normal DNA sources (e.g., cultured fibroblasts) for haematological cancers, and the overall lack of contamination in a class of solid tumours (sarcomas). This analysis could be extended to a broader range of cancer types, especially focusing on those that have begun to spread (such as in late-stage patients) and might present contamination. However, the high incidence and level of contamination observed for other tumour and DNA source types highlights the importance of bioinformatics pipelines that can accommodate TIN contamination. Some variant calling algorithms are now attempting to mitigate the impact of contaminated normal samples[4], but this is more difficult for complex and structural variants for which variant calling from short read data is more challenging and less well established.

The high proportion and high level of contamination observed for some haematological cancer subtypes demonstrates the importance of careful consideration of normal sample source and collection protocols. It is paramount that bioinformatics pipelines are capable of detecting and reporting TIN contamination and subsequently mitigating the impact on somatic variant detection. Thus we recommend that the assessment of tumour-in-normal contamination using a tool such as TINC becomes part of standard quality control procedures for tumour-normal matched pair analyses, especially in clinical settings where data are being used in patient care.

## Methods
### Ethics
Approval for the 100,000 Genomes Genomics England project was obtained from the national research ethics committee (IRAS ID 166046). Participants were selected on the basis of having been identified by healthcare professionals and researchers within the NHS as having a cancer diagnosis. The participants were recruited across 13 NHS Genomic Medicine Centres and written informed consent was obtained from the participants.

### The TINC method
TINC tracks putative clonal somatic SNVs in the tumour and normal samples to determine the overall purity of the tumour (TIT score), and the contamination of tumour cells in the normal biopsy (TIN score). As input it requires the read counts of somatic SNVs (e.g. Strelka2 VCF file) and, if available, the CNA segments (e.g. Canvas VCF file).

TIT and TIN scores are computed as fractions (0 to 1). These can be represented either as the fraction of tumour cells or the fraction of reads originating from tumour DNA in the normal sample. These values are equivalent if the tumour genome is diploid. The conversion

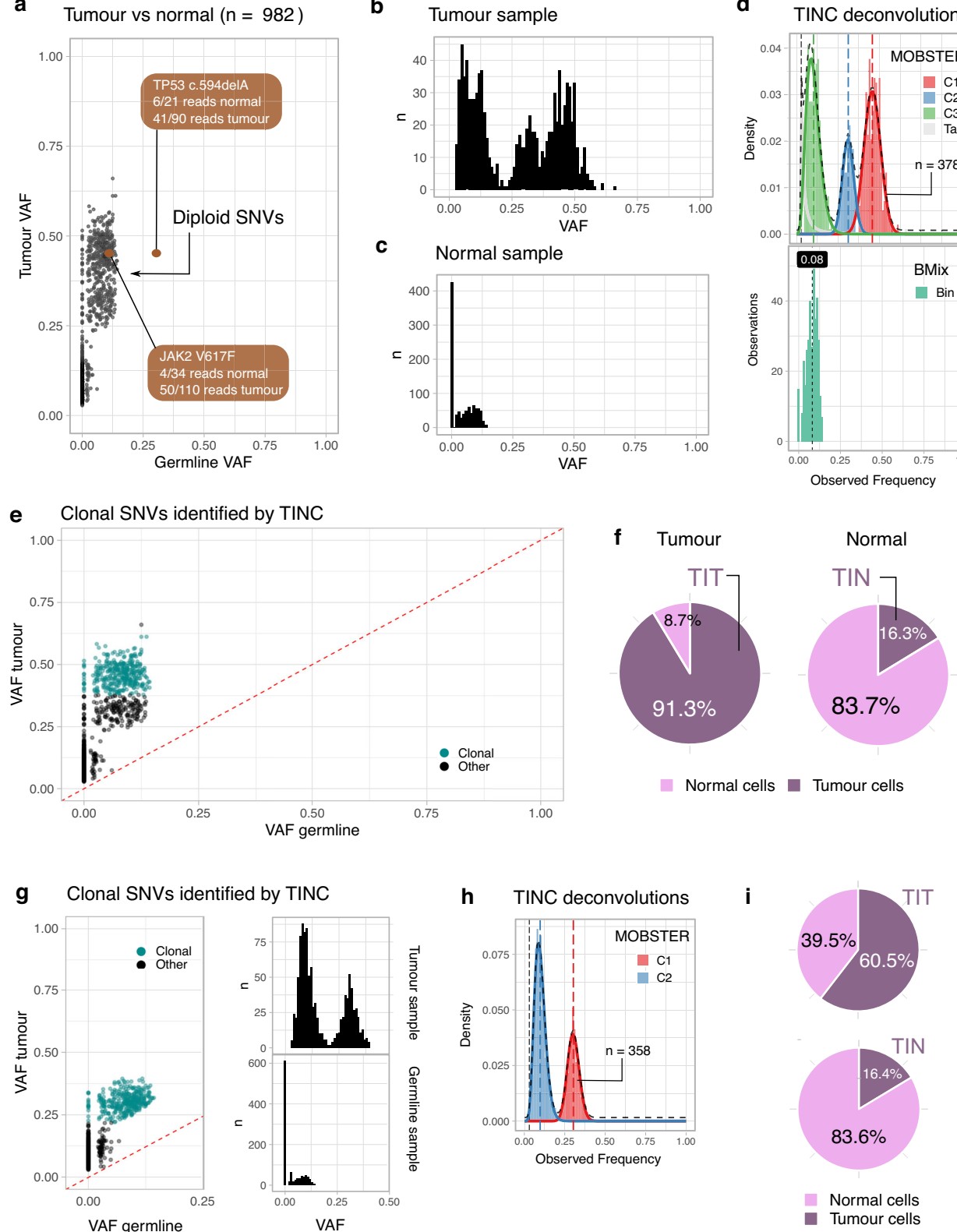

between the two values (cell fraction and read fraction) requires the knowledge of tumour CNAs in order to normalise observed VAF values for the tumour ploidy profile.

**Bulk tumour deconvolution.** The subset of clonal SNVs is derived from the input set of somatic variants using the MOBSTER mixture model[9,24]. For data $x$, the likelihood is that of a $(k+1)$-dimensional

Dirichlet mixture

$$f(x|\theta, \pi) = \pi_1 PL(x|\theta) + \sum_{i=2}^{k} \pi_i Beta(x|\theta) \quad (1)$$

Here $\theta$ are model parameters considering the density function $PL(x|\theta)$ of a Pareto Type-I power law, and $Beta(x|\theta)$ for a Beta

**Fig. 7 | Examples of TINC test outputs. a** Scatter distribution of somatic mutation VAF in tumour and normal samples (**a**–**f** represent case 1). VAF is shown for n=982 mutations detected from WGS data which reside within heterozygous diploid regions in the tumour genome. Two variants of clinical significance are highlighted; a *TP53* frameshift deletion (c.594delA) and a *JAK2* V617F mutation. Neither mutation would be detected using a standard tumour-normal calling pipeline, due to the tumour contamination in the normal. **b, c** Histograms of VAF values for tumour and normal samples in (**a**). **d** Deconvolution analysis with TINC. $n = 378$ clonal mutations were identified in the tumour using MOBSTER (upper panel) with mean VAF ~45% (cluster C1). Subsequent deconvolution determines one cluster in the normal sample for the corresponding mutations with a VAF peak at about ~8% (lower panel). **e** Representation of somatic mutation VAF in tumour and normal samples. After deconvolution of somatic mutations (**d**), clonality can be attributed to the mutations in (**a**)–clonal mutations with teal dots. **f** TIT and TIN scores can be determined from the parameters fit by the deconvolution methods, accounting for the copy state of somatic SNVs. In this case, the data indicate an overall tumour purity of 90% (TIT score, high-purity tumour sample) and tumour-in-normal contamination level of ~16% (TIN score). **g** Representation of somatic mutation VAF in tumour and normal samples (**g**–**i** represent case 2) as in (**a**–**c**). For this case, a previously identified (by Fluorescence in situ hybridisation) translocation resulting in a PML-RARA fusion was not detected using a standard tumour-normal analysis pipeline. **h** Deconvolution identifies a cluster of clonal somatic mutations of $n = 358$ SNVs (cluster C1) with VAF ~30%. **i** Representation of contamination in tumour and normal samples. TIT and TIN scores determined by TINC, expressed in cellular proportions and adjusted for copy number states, show a tumour purity of ~60% (TIT), and tumour contamination of the normal sample of ~16% (TIN).

distribution; $\pi$ are mixing proportions representing the proportion of mutations assigned to each cluster.

The fit is carried out to select the best possible model, using a score function based on the principles of the integrated classification likelihood (ICL), which extends the likelihood $f(x|\theta,\pi)$ with a regularisation for the complexity of the model—measured by $|\theta| + |\pi|$—akin to the Bayesian information criterion, and include the separation of the clusters through an entropy term defined over latent variables $z_{n,k}$ (clustering responsibilities)

$$ICL = -2\log(x|\theta,\pi) + (|\theta| + |\pi|)\log(n) + \Sigma_{z_{n,k}} z_{n,k} \log(z_{n,k}) \quad (2)$$

Through a gradient-based procedure, MOBSTER optimises the value of $k$ and the presence of the tail in the data as defined by the PL density, improving largely over standard methods for subclonal deconvolution[9]. In general, MOBSTER can be used to study the full tumour architecture, a more broad and complicated problem than just determining clonal mutations, as we perform here; nonetheless, the tool is fast and provides precise evolutionary information that TINC can use to estimate contamination.

From MOBSTER fits we obtain a set of vector-valued latent variables $z_{n,k}$ reporting the probability of each input mutation to be part of one of a set of clusters; these contain the normalised posterior densities

$$z_{n,k} = [\pi_k g(x|\theta_k)]/[\Sigma_i \pi_i g(x|\theta_i)] \quad (3)$$

were $g(\cdot)$ are the density functions used in the mixture (power-law and Beta, depending on the component index $i$).

By design MOBSTER labels with "C1" (or $C_1$) the set of mutations with the highest VAF. These should be the clonal mutations we need in TINC, unless there are CNA events in the data and TINC is run without input CNAs. If that is the case, e.g. the tumour bears some miscalled large loss of heterozygosity (LOH), then $C_1$ might be artifactually related to the CNA event. Mistaking such a cluster for clonal mutations would inflate TIT and TIN estimates; therefore TINC maps mutations in $C_1$ to chromosomes, and tests for their enrichment when it is run without CNA data.

Mutations mapping is done by a function $map(\cdot)$ that returns the chromosome counts were a set of mutations map onto the genome. Cluster C1 is rejected—as the putative clonal cluster—by using an empirical 60/20 rule: if more than 60% of mutations in $C_1$ map to less than 20% of the chromosomes (we define this to be the case in which $map(C_1, 0.6) < 0.2$). The 60/20 cutoffs are determined from analysis of pan-cancer WGS data at Genomics England.

When C1 is rejected, TINC performs a recursive test for clusters with progressively lower Beta means, stopping when a suitable cluster is found or all clusters are rejected; in the latter case, TINC determines it is impossible to assess reliable clonal mutations. In practice, the TINC

algorithm selects $C_i$ such that

$$map(C_i, 0.6) > 0.2 \wedge \forall j < i. map(C_j, 0.6) < 0.2 \quad (4)$$

When TINC is run with CNA data this test is not required since we are already filtering mutations by the tumour's most prevalent karyotype. This choice allows control for CNAs that confound the VAF distribution[10].

Note that by running TINC with CNA one elicits the assumption that CNA segments are correct. We also note that, using CNAs for tumour with high friction of CNAs (i.e., cases with copy neutral LOH, triploid or tetraploid genomes etc.), the mutations associated with the cluster with highest VAF are those that happened before the copy number event. In other words, if we work with a tumour that is prevalently triploid with two and one copies of the major and minor alleles, the putative subset of clonal mutations that we find in the high-VAF cluster have happened before the amplification of the major allele.

When TINC has identified a cluster $w$ of putative clonal mutations, it selects the ones with assignment probability above a threshold $z_+ > 0$, disregarding all others. I.e., it defines

$$C = \{n|z_{n,w} > z_+\} \quad (5)$$

If there are not enough such mutations, $z$ is decreased until we include a predefined number of mutations that the user can decide. For instance, TINC can be parameterised to search for all mutations with at least 90% probability to be assigned to the clonal cluster, requiring at least $n = 150$ mutations back. With its dynamic-cutoff strategy, TINC might determine that, in order to select 150 clonal mutations, $z_+$ must be decreased to 80%.

**Bulk normal deconvolution.** Read counts for putative clonal mutations are collected from the normal biopsy, and then used by TINC to fit a Binomial mixture model. This is available in the open-source R package BMix[9], which provides univariate Binomial and Beta-Binomial mixtures. For a set of clonal mutations with associated sequencing depth $d > 0$, and $r \geq 0$ reads with the alternative allele, we use the likelihood

$$f(x = [r, d]|\theta, \pi) = \sum_{i=1}^{k} \pi_i Bin(r|d, \theta) \quad (6)$$

Here we have a mixture of $k \geq 1$ components, where $b$ is a Binomial density function for $d$ trials with $r$ successes, and unknown success probability $p$ (parameter in $\theta$). In this case BMix optimises the value of $k$ scoring models similarly to MOBSTER, and then returns clustering assignments, latent variables, and Binomial parameters $p_1,...,p_k$ similar to MOBSTER[9].

**TIT and TIN scores.** The computation of TIN and TIT scores is done after deconvolution of tumour and normal bulks. In both cases TINC uses the same principle to normalise for the tumour genome

karyotype, which requires knowing copy number data or assuming a known state (e.g., diploid).

We use our CNAqc open-source R package for WGS to make these conversions[10]. For mutations mapping on a genome segment with $m$ copies of the minor allele, $M$ of the major, allelic frequency $v$ and mutation multiplicity $\mu$ (i.e., number of copies of the mutation on the genome), the tumour purity $\rho$ is determined by

$$[\mu + v(2 - m - M)]\rho = 2v \qquad (7)$$

Solving for $\rho$ we determine the percentage of tumour cells in the biopsy.

When TINC is run without CNA data, the model assumes that the above equation is to be solved for a diploid tumour ($m = M = \mu = 1$).

This equation is used for both TIT and TIN estimation, using different values for $v$ in the tumour, and in the normal.

- For TIT, $v$ is the mean VAF of clonal mutations identified by TINC (mapping to cluster $w$ in MOBSTER analysis)

$$v = mean\{VAF_x | x \in w\} \qquad (8)$$

- For TIN, $v$ is expressed as a linear combination of the Binomial peaks and cluster sizes reported by BMix, i.e., a mean of $p_1, ..., p_k$ weighted by the mixing proportions $\pi_1, ..., \pi_k$,

$$v = \sum_{i=1}^{k} \pi_i p_i \qquad (9)$$

For the normal biopsy, we use all the mutations determined in the tumour as clonal (red MRCA in Fig. 1b). One could be tempted to subtract those with VAF above 0 from that set, which are theoretically those accruing from the MRCA of both the cells in the tumour and the normal (yellow MRCA in Fig. 1b). However, we are expecting to work with quite low VAF values corresponding to a normal with contained levels of contamination. Given the median coverage of WGS assays available in Genomics England for normal biopsies (30x), we cannot neglect a strong effect of Binomial sampling (sequencing observational model) on the observed counts. For this reason, retaining all highly-confident clonal mutations in the tumour is a reasonable conservative choice. Their effect on TINC estimation is weighted by their proportions, as obtained from BMix clustering.

TIT and TIN scores can be finally converted to units of read counts by solving for another equation. If we denote with $\lambda$ the tumour cellular fraction, the read counts fraction $\eta$ is

$$\eta = \frac{\lambda(m + M)}{\lambda(m + M) + 2 - 2\lambda} \qquad (10)$$

This equation follows from simple arguments about allele multiplicity[10,25,26].

**Complementary multivariate analysis.** A variational Binomial mixture model is used jointly on read counts data of both the tumour and normal biopsies; this is available through the VIBER open-source R package[9], which is designed to implement multi-dimensional mixtures with arbitrary dimensions.

This type of mixture is semi-parametric and determines the number of clusters in an automatic fashion via variational inference, using the likelihood function

$$f(x|\theta, \pi) = \sum_{i=1}^{k} \pi_i Bin(r_t|d_t, \theta_{i,t}) Bin(r_n|d_n, \theta_{i,n}) \qquad (11)$$

where the counts are considered for both the normal ($d_n$ and $r_n$) and tumour ($d_t$ and $r_t$) assays. Here we are assuming that the counts are independent, as obtained from two sequencing runs.

From VIBER outputs TINC checks the position of the clonal mutations identified for tracking. When these associate a single cluster —as one might expect—we can obtain an alternative TIT and TIN set of scores; TINC uses these to confirm the original estimates obtained by using MOBSTER and BMix, and reports this further evidence to the user. However, we note that results from this joint analysis do not fully count as a joint recalling step of somatic variants, as available by default in DeTiN[8], for instance. This is because the input to this analysis is obtained from the standard paired tumour-normal workflow, and therefore the input itself is affected by the false negative calls. In the future, however, one might think of extending TINC by implementing a straightforward recalling step through a pileup of variant read counts data from both the tumour and normal samples. In this sense, since VIBER outputs are already available inside the TINC object, adding this feature should not be complicated besides the cost of preparing the data by a pileup.

### Comparison with DeTiN

DeTiN[8] was run with both somatic single-nucleotide variants (SSNVs) and allele-specific somatic copy-number alterations (aSCNAs) as input. DeTiN estimates contamination for each of these somatic variant types separately, and then combines them into a single value which we report here.

SSNVs were generated with the Genomics England Pipeline 2.0. We used GATK's (v4.0.4.0) CNA analysis suite (https://github.com/broadinstitute/gatk), utilising a panel of normals, to generate the aSCNAs input required for DeTiN.

### Mapping/variant calling pipeline

**Genomics England Pipeline 2.0 (implemented in November 2020).** All samples were sequenced on HiSeq platform to an average coverage of 100x for tumour and 30x for normal. Read alignment against the human reference genome GRCh38+Decoy+EBV was performed with DRAGEN software (version 3.2.22). Small variant calling together with tumour-normal subtraction was performed using Strelka (version 2.9.9).

In addition to default Strelka filters we applied the following additional filters in order to reduce the false positive rate in the set of somatic variants:

1. Variants with a population germline allele frequency above 1% in the Genomics England dataset of >6000 unrelated individuals or gnomAD v2 datasets;
2. Recurrent somatic variants with a frequency above 5% in the Genomics England dataset;
3. Variants overlapping simple repeats as defined by Tandem Repeats Finder;
4. Small indels in regions with high levels of sequencing noise where at least 10% of the basecalls in a window extending 50 bases to either side of the indel's call have been filtered out by Strelka due to the poor quality;
5. SNVs resulting from systematic mapping and calling of artefacts. We tested whether the ratio of tumour allele depths at each somatic SNV site were significantly different to the ratio of allele depths at this site in a panel of normals (PoN) using Fisher's exact test. The PoN was composed of a cohort of 7000 non-tumour genomes from the Genomics England dataset, and at each genomic site only individuals not carrying the relevant alternate allele were included in the count of allele depths. The mpileup function in bcftools v1.9 was used to count allele depths in the PoN, and to replicate Strelka filters duplicate reads were removed and quality thresholds set at mapping quality > = 5 and base quality > = 5. All somatic SNVs with a Fisher's exact test phred score < 50 were filtered.

Copy number aberrations were identified with Canvas 1.39. Structural variants were identified with Manta (version 1.5). Population germline allele frequency for the breakpoints of a given structural variant is based on two internal panels of normals: GESG, which consists of germline variants coming from single germline analysis of about 2200 samples, and GECG, which consists of the variants detected as germline in paired tumour-normal variant calling for about 2,500 cancer samples. If a variant has two breakpoints, maximal value of allele frequency among the two is reported.

**Genomics England Pipeline 2.0 for samples with TIN contamination.** Haematological samples with TIN > 1% or where TIN can't be estimated (e.g. due to TIT content < 25%) are also analysed in the parallel pipeline run without subtracting variants from the patient's germline. Gender-matched platinum genome is used as a normal sample to satisfy input requirements for Strelka and Manta. Subsequent filtering of variants with population frequency >0.01 and variants highlighted with PoN (see above) significantly reduces contamination by unsubtracted germline variants and sequencing artefacts. The results of two pipeline runs are subsequently merged and analysed together in the annotation and interpretation workflow. In the WGS analysis results high-confidence somatic variants from the paired tumour-normal pipeline are highlighted with SOMATIC flag while the outcomes of tumour only pipeline that pass filters are presented as variants of UNCERTAIN origin.

**In silico contamination data generation**
Tumour-in-normal contamination was generated in silico for cohorts of haematological and lung cancer tumour-normal pairs, with samples selected based on a Ccube tumour purity estimate > 30% and baseline normal samples checked to ensure they were not contaminated[27]. Various levels of tumour-in-normal contamination were created for each tumour-normal pair by using samtools to combine fractions of the normal and tumour BAM files. The level of contamination was calculated accounting for the purity of the tumour sample predetermined with Ccube[27]. Using the in silico contaminated normal bam files TINC R package input was generated using the Genomics England pipeline outlined above.

**Fluorescence-activated cell sorting (FACS) for quantifying MRD**
A sequential gating strategy is applied at diagnosis to establish and define patients leukaemia-associated aberrant phenotype (LAIP) using comprehensive 8 colour panels (8–9 antibodies for B-ALL) and BD FACS Canto II instrument. Data is analysed using FACS DIVA software (BD). Once established, the same sequential LAIP gating is applied to all follow up samples to quantify MRD. MRD events found in the final LAIP gate are reported as a percentage divided by the total number of CD45 positive "live" WBC events analysed. A "different from normal" approach is also utilised when LAIP is similar to normal BM haematopoiesis or where there is marked regeneration in LAIP gates at later time points.

Standard Flow MRD Panels include:
B-ALL MRD
CD19, CD20, CD66c/CD123, CD38, CD10, CD45, CD34, CD81
CD19, CD20, CD73/CD304, CD38, CD10, CD45, CD34, CD81
AML MRD
CD45, CD34, CD13, CD33, HLA DR, CD117, CD11b, CD15
CD45, CD34, CD56, CD33, CD38, CD117, CD11b, CD15
T-ALL MRD
CD56/TCR alpha/beta/TCR gamma/delta (cocktail), CD99, CD45, CD5, CD7,CD3, CD4, CD8

CD56/TCR alpha/beta/TCR gamma/delta (cocktail), CD2, CD45, CD5, CD7,CD3, CD4, CD8

**Analysis of patients data**
**Samples and data collection.** All samples as well as sample metadata were collected as part of Genomics England 100,000 Genomes Project[28]. Sample Specifications for haematological cancers are described in Supplementary Table 1.

**Identification of potential CHIP variants.** The list of 168 CHIP-associated SNVs was compiled from the literature[15], gathering pathogenic variants in the genes known to drive CHIP and myeloid malignancies and identified in at least three cases in the cohort of 46,706 unrelated healthy individuals. For example, variants in the following genes were included: *IDH2, PRPF8, PPM1D, SRSF2, TP53, GNB1, ASXL1, GNAS, RUNX1, SF3B1, DNMT3A, MYD88, CCND3, TET2* and *JAK2*. Genomic data from the normal samples of 168 AML patients was scanned with *bcftools mpileup* to calculate support for CHIP-associated variants. In order to examine relationships between CHIP and contamination estimated by TINC, VAF of CHIP-associated mutation in normal samples was correlated with TIN score (in Read Fraction units with SNVs and CNAs used in calculation).

**Reporting summary**
Further information on research design is available in the Nature Portfolio Reporting Summary linked to this article.

## Data availability
TINC results used to produce figures in the manuscript are provided as a Source Data file, in Excel format with multiple sheets and anonymised sample IDs. Original sample IDs are available with a copy of the data stored in the Genomics England Research Environment, in the "/published_data_archive/paper_data/paper_data_RR306" folder. The description of the data available in the Genomics England Research Environment for this paper will be available under https://re-docs.genomicsengland.co.uk/tinc_publication/. The sequencing data and variant calls supporting the findings of this study are available within the Genomics England Research Environment, a secure cloud workspace. To access genomic and clinical data within this Research Environment, researchers must first apply to become a member of either the Genomics England Clinical Interpretation Partnership, GECIP (https://www.genomicsengland.co.uk/research/academic) or the Discovery Forum (industry partners https://www.genomicsengland.co.uk/research/research-environment). The process for joining the GECIP is described at https://www.genomicsengland.co.uk/research/academic/join-gecip and consists of the following steps: Your institution will need to sign a participation agreement available at https://files.genomicsengland.co.uk/documents/Genomics-England-GeCIP-Participation-Agreement-v2.0.pdf and email the signed version to gecip-help@genomicsengland.co.uk. Once you have confirmed your institution is registered and have found a GECIP domain of interest, you can apply through the online form at https://www.genomicsengland.co.uk/research/academic/join-gecip. Once your Research Portal account is created you will be able to log in and track your application.The domain lead will review your application within 10 working days. Your institution will validate your affiliation. You will complete our online Information Governance training and will be granted access to the Research Environment within 2 hours of passing the online training. Source data are provided with this paper.

## Code availability
TINC (https://github.com/caravagnalab/TINC/) is available as an open source R package hosted at GitHub, and the release used for this

manuscript is available at a Zenodo repository[29]. The TINC website at https://caravagnalab.github.io/TINC/ presents detailed manuals and RMarkdown vignettes for all TINC analyses. The source code to replicate all the figures of this paper is available within the Genomics England Research environment (under the "/published_data_archive/paper_data/paper_data_RR306" folder). A copy of this data with anonymised sample identifiers is also available in a Zenodo repository[30].

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

## Acknowledgements

This research was made possible through access to data in the National Genomic Research Library, which is managed by Genomics England Limited (a wholly owned company of the Department of Health and Social Care). The National Genomic Research Library holds data provided by patients and collected by the NHS as part of their care and data collected as part of their participation in research. The National Genomic Research Library is funded by the National Institute for Health Research and NHS England. The Wellcome Trust, Cancer Research UK and the Medical Research Council have also funded research infrastructure.

The research leading to these results has received funding from AIRC under MFAG 2020—ID. 24913 project—P.I. Caravagna Giulio.

We wish to thank Jude Fitzgibbon and Trevor A Graham for helpful discussions on a preliminary version of this manuscript.

## Author contributions

G.C., J.M. and A.S. conceived TINC, which G.C. formalised and implemented with support from S.M. J.M. and M.Z. ran all analyses and synthetic tests; J.M., S.M., L.V., G.C. and A.S. interpreted the results. N.V. and O.Y. implemented hybrid variant calling for processing samples with TIN contamination in Genomics England production pipeline and provided accuracy estimates. J.C., R.T. and J.B. performed experimental validation. G.C., A.S., S.W., J.B. and J.M. drafted the manuscript, which all authors finalised and approved in its final version.

## Competing interests

The authors declare no competing interests.
