## [Peer Review File · Nature Communications]

REVIEWER COMMENTS

Reviewer #1 (Remarks to the Author): Expert in bioinformatics, cancer genomics and evolution, and sarcomas

Mitchell et al. propose a new method (TINC) to infer the tumour contaminant level in normal germline proxy (TiN) from the allele frequencies of somatic single nucleotide variants (SSNVs) in both samples in high-throughput sequencing experiments.

This is an important part of somatic mutation analyses, as the normal germline proxy is used to remove non-somatic variants; and the presence of tumour cells in the normal would lead to over-filtering of (potentially clinically relevant) mutations.

So far only one tool has been available for this: DeTiN. The authors mention two improvements over DeTiN: 1) Unlike DeTiN, TINC infers both the tumour in normal and the tumour in tumour (TIT) simultaneously. Thus TINC can also directly be used to infer minimal residual disease (MRD). 2) TINC is compatible with any SNV and CNA callers whereas DeTiN depends on the outputs of specific tools.

Similarly to what was done for DeTiN in the original publication, the authors of TINC employ in-silico merging of tumour and normal reads in known proportions to evaluate the accuracy of their tool. They also apply them to germline-proxy samples of malignancies for which this baseline is known to likely be contaminated.

Overall, their approach reaches good accuracy and correctly classifies samples with minimal residual disease, and agrees with independent measurements by flow cytometry or molecular testing.

Altogether, this new approach is an interesting tool for cancer genomics analyses, which answers a well known problem for mutation calling in the tumour-normal setting. The method is well described and validated against sound in-silico simulated ground truth.

TINC is a tool for TiN estimation that matches the accuracy of the previous tool, while providing wide compatibility with upstream callers. However, it seems to me that DeTiN is a better tool than what the authors make of it here: the authors seem to overlook major advantages of DeTiN over TINC, and to overinflate one of the apparent disadvantage (estimation of the ratio rather than TiN). Thus, apart from

the ease of use of TINC, which is a valid point, I am not sure what the advantages of TINC over a user-friendly wrapper around DeTiN really are.

I would have a few related comments that I think the authors might want to address to convince that TINC is indeed a good alternative over the one already-existing tool (or a wrapper around it).

Major comments.

1a) the only published methodology for TiN estimation and thus the default gold standard, DeTiN, uses both the SSNVs and the SCNAs with SNP phasing to infer TiN. This has an obvious advantage over TINC, which only uses SSNVs - even though SCNAs are processed for filtering SNVs, they are not used in the actual inference of TiN. SCNAs are present in most cancers and thanks to the phasing of SNPs, the power to detect minimal levels of TiN from SCNA is high. This has been shown to study mosaicism in blood [e.g. <https://doi.org/10.1038/s41586-018-0321-x>] and could be used to distinguish mosaicism from tumour in normal. SCNA-based TiN inference would be especially useful for highly aberrated samples, for which TINC would not be able to include CNAs information in the SSNV-based inference of TiN. This should be better discussed by the authors.

1b) Related to this, could the authors please comment on the 7.5% of sarcomas not analysed on Supplementary Figure S5. This seems like a large number but it is unclear why they were not analysed. If due to the limitations of TINC, this should be emphasised.

1c) The authors should describe in which mode they have run DeTiN (especially using CNA and SNP phasing or not?) - this is absent from the methods; and explore the effect of the use of SCNAs for TiN estimation in their simulations - so they can hopefully show that TINC is as good as DeTiN despite not using SCNAs-based TiN inference.

2a) Another big difference is in somatic mutation calling after TiN estimation: DeTiN proposes a way to rescue candidate SSNVs after TiN estimation; what about TINC? Should the samples simply be processed without matched normal (or even discarded?) or could the normal be used in a similar fashion to DeTiN? This seems a particularly good usage of VIBER in TiN vs. TIN space?

2b) The authors could benchmark rescuing variants in Strelka calls from runs on tumour-normal pairs where the original normal (with TiN estimate < 1%) has been spiked in with tumour contaminant vs. the Strelka calls on the original tumour-normal pair.

3) The authors claim that the advantage over DeTiN is that TINC estimates both the TiN and the TiT whereas DeTiN estimates the ratio. While this is indeed what DeTiN estimates, the authors of DeTiN do propose that TiN can trivially be inferred from the ratio, given sample purity and ploidy, which are standard outputs of copy-number callers. This is thus not a real advantage of TINC over DeTiN. Also, the authors could compare these estimates with the estimates of TINC directly.

4) Given the above three points on DeTiN (SCNA-based inference of TiN; rescuing of SNVs; estimation of TiN/MRD), what is the advantage of TINC over making DeTiN more widely compatible, e.g. having a wrapper around DeTiN to make it compatible with outputs from other callers?

5) Does the presence of hotspot mutations in the blood always correspond to the detection of TiN by TINC? What about variants with no evidence in the tumour but found in the normal? Could the authors further comment on the potential presence of hotspot mutations as mosaicism in the blood rather than as evidence of TiN (such as the TP53 example) and how these could be distinguished from tumour mutations from TiN contaminant?

Minor comments.

Could the authors describe what type of fresh frozen "tissues" were used as germline proxy for two of the contaminated sarcoma samples? The use of "tissue" as opposed to blood is ambiguous, since blood is a type of tissue.

Figure S7F, S8F, S10F: do the colours match the labels? C1 should be the clonal cluster?

Reviewer #2 (Remarks to the Author): Expert in cancer genomics, evolution, and bioinformatics

Caravagna and colleagues present a manuscript introducing TINC, a new computational tool to assess tumor-in-normal contamination (TiN) in WGS data. This is an important issue especially for hematological malignancies as the matched normal samples are more likely to be contaminated by malignant cells. The authors assessed the performance of their tool in simulated data created from tumor/normal pairs from hematological and solid cancers as well as in an orthogonal dataset of minimal residual disease from leukemia patients and then apply it to a large cohort of tumors from the Genomic England 100,000 Genomes Project.

Similar to DeTIN, TINC uses somatic SNV calls from matched tumor/normal analysis and assesses the presentation of these in the tumor and normal samples. However, in TINC the focus is on clonal SNVs as identified using MOBSTER. Moreover, different from DeTIN, TINC can also take into account allele-specific CNA calls to filter for SNVs falling into the CNA segments with the most prevalent CN state to avoid bias due to changes in ploidy. This is important as local ploidy of somatic SNVs will confound the TIN estimates. The authors clearly demonstrate this in the analysis of the in silico contaminated normals from lung cancer patients.

Overall, this is a well-written manuscript that presents a computational tool of interest to the genomic community. I have no further comments.

Response to Reviewers

Reviewer 1

Reviewer #1 (Remarks to the Author): Expert in bioinformatics, cancer genomics and evolution, and sarcomas

Mitchell et al. propose a new method (TINC) to infer the tumour contaminant level in normal germline proxy (TIN) from the allele frequencies of somatic single nucleotide variants (SSNVs) in both samples in high-throughput sequencing experiments.

This is an important part of somatic mutation analyses, as the normal germline proxy is used to remove non-somatic variants; and the presence of tumour cells in the normal would lead to over-filtering of (potentially clinically relevant) mutations.

So far only one tool has been available for this: DeTiN. The authors mention two improvements over DeTiN: 1) Unlike DeTiN, TINC infers both the tumour in normal and the tumour in tumour (TIT) simultaneously. Thus TINC can also directly be used to infer minimal residual disease (MRD). 2) TINC is compatible with any SNV and CNA callers whereas DeTiN depends on the outputs of specific tools.

Similarly to what was done for DeTiN in the original publication, the authors of TINC employ in-silico merging of tumour and normal reads in known proportions to evaluate the accuracy of their tool. They also apply them to germline-proxy samples of malignancies for which this baseline is known to likely be contaminated.

Overall, their approach reaches good accuracy and correctly classifies samples with minimal residual disease, and agrees with independent measurements by flow cytometry or molecular testing.

Altogether, this new approach is an interesting tool for cancer genomics analyses, which answers a well known problem for mutation calling in the tumour-normal setting. The method is well described and validated against sound in-silico simulated ground truth.

We thank the reviewer for the positive assessment of our work, and for appreciating our effort to integrate in-silico testing as well independent experimental validation for TINC.

TINC is a tool for TIN estimation that matches the accuracy of the previous tool, while providing wide compatibility with upstream callers. However, it seems to me that DeTiN is a better tool than what the authors make of it here: the authors seem to overlook major

advantages of DeTiN over TINC, and to overinflate one of the apparent disadvantage (estimation of the ratio rather than TiN). Thus, apart from the ease of use of TINC, which is a valid point, I am not sure what the advantages of TINC over a user-friendly wrapper around DeTiN really are.

We appreciate your feedback and the insights you have provided, and acknowledge that DeTiN is a valuable tool and we understand your perspective regarding its advantages over TINC. While we agree that DeTiN offers certain benefits, we would like to emphasise that our intention was not to downplay its capabilities or overlook its advantages. Our goal in this work was to present TINC as a valid alternative with its own set of advantages, especially in terms of wide compatibility with upstream callers and ease of use.

TINC is a general solution offering compatibility with various upstream callers, which allows researchers and practitioners to seamlessly integrate this method into existing workflows, reducing the need for significant modifications or customizations. We understand your point regarding a user-friendly wrapper around DeTiN, which could potentially provide similar benefits. However, we believe that TINC's broader compatibility and lightweight nature make it a valuable alternative for researchers and practitioners who require a more general solution. We discuss this more in detail in response to your point 4 (below).

Once again, we appreciate your thoughtful review and the opportunity to clarify our perspective. We will certainly take your comments into consideration for future improvements and discussions. We reply below to your major comments.

I would have a few related comments that I think the authors might want to address to convince that TINC is indeed a good alternative over the one already-existing tool (or a wrapper around it).

Major comments.

1a) the only published methodology for TiN estimation and thus the default gold standard, DeTiN, uses both the SSNVs and the SCNAs with SNP phasing to infer TiN. This has an obvious advantage over TINC, which only uses SSNVs - even though SCNAs are processed for filtering SNVs, they are not used in the actual inference of TiN. SCNAs are present in most cancers and thanks to the phasing of SNPs, the power to detect minimal levels of TiN from SCNA is high. This has been shown to study mosaicism in blood [e.g. <https://doi.org/10.1038/s41586-018-0321-x>] and could be used to distinguish mosaicism from tumour in normal. SCNA-based TiN inference would be especially useful for highly aberrated samples, for which TINC would not be able to include CNAs information in the SSNV-based inference of TiN. This should be better discussed by the authors.

TINC uses SCNAs to select a subset of SNVs in genomic intervals with the copy number state spanning the largest proportion of the tumour genome, It then adjusts variant supporting read counts according to allele-specific CNA configuration to compute TIN level. While simpler than DeTIN, our model also incorporates copy number data.

Regarding general levels of aneuploidy in highly-aberrated samples, we now report the CNA profiles (whole-genome allele-specific segmentation) of the lung samples we used for testing TINC, visualised using the CNAqc package for joint analysis of CNAs and somatic mutations data (<https://caravagnalab.github.io/CNAqc/>). These samples show a large degree of aneuploidy (average 78% of the genome affected by CNAs, per sample), as we report now in a new Supplementary Figure S2. TINC and DeTIN have shown very similar results for this cohort (Figure 1e) with correlation between expected and predicted values $R^2=0.9$ for TINC and $R^2=0.92$ for DeTIN. Advantages of using CNAs in TINC estimates are discussed in “In silico validation of TINC performance” section and demonstrated by comparison of Figure 1b,c (TINC runs with CNA) and Supplementary Figure S1a,b (TINC runs without CNA). Therefore we confidently conclude “*these data demonstrated the benefit of incorporating CNAs, particularly for cancers with high chromosomal instability*”, in agreement with Reviewer’s comment.

1b) Related to this, could the authors please comment on the 7.5% of sarcomas not analysed on Supplementary Figure S5. This seems like a large number but it is unclear why they were not analysed. If due to the limitations of TINC, this should be emphasised.

A subset of sarcoma samples was not analysed on Supplementary Figure S6 due to the low tumour purity, We added the following sentence to the legend for Supplementary Figures S5 and S6 : “*The proportion of cases that could not be analysed by Genomics England pipeline (tumour purity estimated to be below 25%) is shown in grey*”. Motivation for this cut-off is presented on Figure 5 covering implementation of TINC pipeline at Genomics England.

1c) The authors should describe in which mode they have run DeTIN (especially using CNA and SNP phasing or not?) - this is absent from the methods; and explore the effect of the use of SCNAs for TiN estimation in their simulations - so they can hopefully show that TINC is as good as DeTiN despite not using SCNAs-based TiN inference.

The reviewer is correct in pointing out that we have forgotten to report the setup of DeTIN for the comparison against TINC. We apologise and clarify that we have run DeTIN with input copy number data as required for best results, and we have added the missing explanations in the section *Comparison with DeTIN (Online Methods)*.

2a) Another big difference is in somatic mutation calling after TiN estimation: DeTiN proposes a way to rescue candidate SSNVs after TiN estimation; what about TINC? Should the samples simply be processed without matched normal (or even discarded?) or could the normal be used in a similar fashion to DeTiN? This seems a particularly good usage of VIBER in TiT vs. TiN space?

2b) The authors could benchmark rescuing variants in Strelka calls from runs on tumour-normal pairs where the original normal (with TiN estimate < 1%) has been spiked in with tumour contaminant vs. the Strelka calls on the original tumour-normal pair.

We thank the reviewer for these interesting comments, which we address together. Compared to DeTiN our approach does not engage in mutation re-calling. We however clarify that, in the Genomics England pipeline, every sample flagged as contaminated by TINC undergoes a secondary calling tumour only pipeline.

We have now discussed this hybrid pipeline in the sections *Genomics England Pipeline 2.0 for samples with TINC (Online Methods)* and *TINC implementation in a high throughput bioinformatics pipeline (Main Text)*, and added a new Main Text Figure 5 reporting the performance of our tumour-only pipeline. In brief, due to advanced filters implemented in Genomics England, we were able to optimise the specificity of variant calling (for small and structural variants) in clinically-relevant genes without compromising sensitivity.

Regarding the re-calling possibility with VIBER (<https://caravagnalab.github.io/VIBER/>) in tumour-normal space, we thank the reviewer for the suggestion and agree that this would be an interesting avenue to explore in a future extension of TINC. In this sense, however, we note that the current input set of mutations for TINC assessment is created from the standard tumour-normal pipeline, and therefore the current input - by construction - may not contain mutations missed by the tumour-normal pipeline. Therefore, re-calling would require raw data (e.g., BAM files) to prepare augmented inputs for VIBER. At the moment, results from VIBER are stored inside the TINC output object, making it straightforward to use those fits where appropriate; we made this consideration explicit in *Complementary multivariate analysis (Online Methods)*.

3) The authors claim that the advantage over DeTiN is that TINC estimates both the TiN and the TiT whereas DeTiN estimates the ratio. While this is indeed what DeTiN estimates, the authors of DeTiN do propose that TiN can trivially be inferred from the ratio, given sample purity and ploidy, which are standard outputs of copy-number callers. This is thus not a real advantage of TINC over DeTiN. Also, the authors could compare these estimates with the estimates of TINC directly

We thank the reviewer for this comment. We are aware that absolute TiN, with DeTiN, can be inferred if one fixes the tumour purity (TiT) and reverses the ratio TiN/TiT. However, as the reviewer recognises, this requires DeTiN to receive in input purity and ploidy from copy number calling. This input is external to DeTiN and therefore, we feel it is correct to stress that our tool, compared to DeTiN, is able to identify the required TiN/TiT values in a completely autonomous way. We have rephrased the section *In silico validation of TINC performance (Main Text)*.

4) Given the above three points on DeTiN (SCNA-based inference of TiN; rescuing of SNVs; estimation of TiN/MRD), what is the advantage of TINC over making DeTiN more widely compatible, e.g. having a wrapper around DeTiN to make it compatible with outputs from other callers?

The TINC method we developed is specifically tailored to fit both the technology of choice at Genomics England, which is whole-genome sequencing (WGS), and the general bioinformatics pipelines implemented, which go beyond GATK.

Statistical signals in WGS allow for the clonal deconvolution of the tumour composition at a resolution that was not possible at the time DeTiN was developed. We specifically refer to recent developments that include population genetics and machine learning algorithms to perform a high-resolution deconvolution of tumour population from a bulk sample (e.g., PMID 32879509). Moreover, the continuous increase in WGS data quality and sequencing coverage will further push WGS to be the technology of choice for many scenarios, and therefore we saw this as an opportunity to use principles that are different from the ones available in DeTiN and develop a technology-specific solution.

The possibility of wrapping DeTiN around the callers available at Genomics England was also considered in a preliminary phase of this project. To this extent, we sought support to create the required input for DeTiN from Canvas/Dragen calls (the CNA-caller in house at Genomics England); we did this by opening on May 30 2022 a GitHub issue at DeTiN author's page

<https://github.com/getzlab/deTiN/issues/38>

which has remained unanswered. We concluded that although theoretically possible, wrapping any caller around DeTiN might not be trivial, and eventually opted for a more flexible solution that consisted in developing a technology that is 100% independent of the mutation or CNA calling algorithms one might adopt.

Indeed, TINC can work with read-counts data from any mutation caller, and similarly with results from any allele-specific CNA calling algorithm. We felt that this large versatility could be an interesting feature also considering the speed at which new calling algorithms are developed as sequencing technology progresses with more complex developments.

5) Does the presence of hotspot mutations in the blood always correspond to the detection of TiN by TINC? What about variants with no evidence in the tumour but found in the normal? Could the authors further comment on the potential presence of hotspot mutations as mosaicism in the blood rather than as evidence of TiN (such as the TP53 example) and how these could be distinguished from tumour mutations from TiN contaminant?

We thank the reviewer for this insightful comment, which helped us deepen our understanding of the relation between Clonal Hematopoiesis of Indeterminate Potential (CHIP), the manifestation of blood mosaicism in AML patients, and contamination.

To answer this comment, we have carried out additional analyses and assessed whether mosaicism in the blood and contamination are correlated phenomena. We demonstrated that TINC was able to flag normal samples with recurrent CHIP mutations, which is now in a new Supplementary Figure S12. While CHIP is not an instance of contamination per se due to a lineage relationship between the hematopoietic and the AML clones, its effect is still to subtract genuine variants (the CHIP ones) in a standard tumour-normal design. Since we cannot compromise sensitivity of the pipeline that is used for clinical reporting, flagging CHIP samples with TINC is a key step that triggers hybrid pipeline that includes tumour-only run in order to report the true extent of tumour mutations, including CHIP.

We thank again this reviewer for motivating us to gain a better understanding of the relationship between CHIP and TINC. We in fact found the exploration of CHIP variants with no evidence in the tumour very interesting but we think it's tangential to this manuscript and will be better pursued in the separate work.

Minor comments.

Could the authors describe what type of fresh frozen "tissues" were used as germline proxy for two of the contaminated sarcoma samples? The use of "tissue" as opposed to blood is ambiguous, since blood is a type of tissue.

We clarified in the text that it was muscle tissue.

Figure S7F, S8F, S10F: do the colours match the labels? C1 should be the clonal cluster?

Panel F reports an independent clustering run carried out with VIBER. Clusters are labelled by VIBER independently of the cluster labels defined by MOBSTER analysis and reported in panel C of the same figures. The two clustering runs may return different sets of clusters; for this reason the two sets of labels are independent. We made this clear in the caption of Supplementary Figures S9 S12.

Reviewer 2

Reviewer #2 (Remarks to the Author): Expert in cancer genomics, evolution, and bioinformatics

Caravagna and colleagues present a manuscript introducing TINC, a new computational tool to assess tumor-in-normal contamination (TIN) in WGS data. This is an important issue especially for hematological malignancies as the matched normal samples are more likely to be contaminated by malignant cells. The authors assessed the performance of their tool in simulated data created from tumor/normal pairs from hematological and solid cancers as well as in an orthogonal dataset of minimal residual disease from leukemia patients and then apply it to a large cohort of tumors from the Genomic England 100,000 Genomes Project.

Similar to DeTIN, TINC uses somatic SNV calls from matched tumor/normal analysis and assesses the presentation of these in the tumor and normal samples. However, in TINC the focus is on clonal SNVs as identified using MOBSTER. Moreover, different from DeTIN, TINC can also take into account allele-specific CNA calls to filter for SNVs falling into the CNA segments with the most prevalent CN state to avoid bias due to changes in ploidy. This is important as local ploidy of somatic SNVs will confound the TIN estimates. The authors clearly demonstrate this in the analysis of the in silico contaminated normals from lung cancer patients.

Overall, this is a well-written manuscript that presents a computational tool of interest to the genomic community. I have no further comments.

We do thank the review for the positive assessment of our manuscript.

REVIEWERS' COMMENTS

Reviewer #1 (Remarks to the Author):

I thank the authors for their replies, which have answered all my comments. This is an important contribution that elegantly tackles a well known problem.